# Conserved glucokinase regulation in zebrafish confirms therapeutic utility for pharmacologic modulation in diabetes
Nicole Schmitner [1] ✉, Sophie Thumer[1], Dominik Regele[1], Elena Mayer[1], Ines Bergerweiss[1], Christian Helker[2,3], Didier Y. R. Stainier [2], Dirk Meyer [1] & Robin A. Kimmel [1]

Glucokinase (GCK) is an essential enzyme for blood glucose homeostasis. Because of its importance in glucose metabolism, GCK is considered an attractive target for the development of antidiabetic drugs. However, a viable therapeutic agent has still to emerge, prompting efforts to improve understanding of the complex regulation and biological effects of GCK. Using the vertebrate organism zebrafish, an attractive model to study metabolic diseases and pharmacological responses, we dissected the complexities of *gck* regulation and unraveled effects of Gck modulation. We found that while *gck* expression in zebrafish islet cells is constitutive, *gck* expression in the liver is regulated by nutritional status, confirming similarity to the mammalian system. A combination of transgenic *gck* reporter lines and our diabetes model, the *pdx1* mutant, allowed monitoring of *gck* expression under pathological conditions, revealing reduced *gck* expression and activity in the liver, which was unresponsive to nutrient stimulation, and decreased expression in the islet due to the reduced number of β-cells. Gck activation substantially ameliorated hyperglycemia in *pdx1* mutants, without inducing oxidative stress responses in liver or islet. In-depth characterization of Gck activity and regulation at the cellular level in a whole-organism diabetes model clarifies its applicability as a drug target for therapies.

Diabetes mellitus is a heterogeneous group of disorders affecting glucose metabolism that are characterized by elevated blood glucose levels[1]. A key enzyme tightly controlling glucose homeostasis is Glucokinase (GCK), which acts as an insulin-independent glucose sensor for glucose-stimulated insulin release in β-cells[2,3], and regulates glucose uptake and storage in the liver in an insulin- and food-inducible manner[4]. GCK, also known as hexokinase IV, is one of four members of the hexokinase family. Features distinguishing GCK from hexokinase I-III include a low affinity for glucose, sigmoidal kinetics and the lack of inhibition by glucose-6-phosphate[5,6]. In mammals, two organ specific promoters control the expression of *GCK*[7]. Differential splicing results in hepatic and endocrine isoforms that are functionally indistinguishable, although there are differences in amino acid sequence at the N-terminal ends of the *GCK* isoforms[8].

Because of its importance for glucose metabolism and disease, GCK is a promising target for the development of novel antidiabetic drugs[9,10]. Candidate GCK activators (GKAs) were discontinued because of transient efficacy and safety issues[11]. The transient effect of GKAs in clinical trials has been attributed to enhanced β-cell activation and stress, causing a progressive decrease in β-cell mass, even thereby contributing to disease progression. Alternatively, repression of liver GCK by the elevated glucose levels of diabetes may critically dampen the effect of GKAs[12]. In current clinical trials, the activator Dorzagliatin is showing promise[13], but uncertainties remain as to whether treatment primarily acts through β-cell or liver activity modulation, and whether treatment is associated with liver and β-cell stress[14].

A contrasting view suggests that GCK inhibition, by decreasing glycolysis and promoting 'β-cell rest,' may produce benefits in β-cell survival and function, and thus glucose control, in diabetes patients[11,14,15]. The hexokinase inhibitor D-Mannoheptulose has been shown to prevent damaging effects of hyperglycemia in β-cells and islets in vitro[14]; however beneficial effects of glucokinase inhibitors (GKI) on whole diabetic organisms have not been confirmed. A viable therapeutic agent targeting GCK has still to emerge, and therefore considerable efforts are made to characterize the complex biochemical processes regulating its activity[16,17].

Zebrafish is an attractive model system to study the pathogenesis and course of complicated metabolic disorders such as diabetes, and to discover

[1]Institute of Molecular Biology, Center for Molecular Biosciences Innsbruck (CMBI), University of Innsbruck, Innsbruck, Austria. [2]Max Planck Institute for Heart and Lung Research, Bad Nauheim, Germany. [3]Present address: Philipps-University Marburg, Marburg, Germany. ✉e-mail: nicole.schmitner@uibk.ac.at

and characterize new diagnostic and therapeutic targets. Importantly, the cornerstones of mammalian glucose homeostasis are conserved in zebrafish[18–21]. Furthermore, mutations in genes affecting β-cell development in zebrafish result in phenotypes mimicking those associated with human diseases; for example, *PDX1* is linked to genetic forms of diabetes and is associated with increased susceptibility to type 2 diabetes in humans[22] and *pdx1* mutant fish display key features of diabetes, including hyperglycemia, reduced insulin and reduced β-cell number[23].

Although zebrafish offer many possibilities to study the role and potential targetability of Gck, it has not been well described in zebrafish. Expression of an ortholog of the mammalian liver *GCK* was detected in zebrafish liver[24], and its expression and activity were shown to be nutrient dependent[25]. Here we define in detail zebrafish *gck* isoforms that are differentially expressed in metabolic tissues, and we use transgenic reporters to characterize *gck* regulation at the whole-organism level. By combining the reporter lines with our *pdx1* mutant diabetes model[23], we examined *gck* regulation in pancreatic islet versus liver in response to feeding and found disrupted *gck* responses under pathological conditions. We could furthermore determine that pharmacologic Gck activation ameliorates hyperglycemia in *pdx1* diabetic mutants without causing stress in endocrine β-cells or liver. This work lays the foundation for further studies using the zebrafish to identify and evaluate novel therapeutic compounds modulating Gck for the treatment of diabetes.

## Results

### Enhanced genome annotation reveals an additional *gck* isoform

The Ensembl database (Danio rerio version GRCz11) has two protein-coding transcripts (splice variants) annotated to the *gck* gene. The longer isoform, *gck202*, consists of 2495 bp, while *gck201* spans 1427 bp, has a shorter 5' UTR and lacks three 3' exons. To further explore the *gck* locus, we examined the transcriptome annotation of Lawson et al.[26]. This enhanced annotation provides more comprehensive gene models as compared to Ensembl and RefSeq transcriptomes, including critical information about introns, splice sites and UTRs[27]. In contrast to Ensembl annotations, the Lawson data indicates a third isoform of *gck* (referred to as *gck iso3*), which has a more distal first exon (7.4 kb upstream from the first exon of *gck202*) (Fig. 1a). With this modification incorporated, gene and isoform level quantification was performed using published datasets of sorted adult zebrafish pancreatic cells[28] and embryonic hepatocytes[29]. After visual inspection of the alignments, it was apparent that the transcription start site of the transcript *gck iso3* needed to be shifted 613 bp upstream (from chr8:40469379 to chr8:40468766).

Quantification of overall *gck* RNA counts in zebrafish islet[28] showed that β-cells and δ-cells expressed *gck* in the pancreatic islet, while few α-cells expressed *gck* (Fig. 1b). Quantification of overall *gck* RNA counts in zebrafish embryonic hepatocytes[29] indicated that expression initiated at 4 dpf and dropped to undetectable levels at 6 dpf (Fig. 1b). As liver expression of *gck* is reported to be nutrient dependent[24,30], we hypothesize that this drop in expression is due to the switch from internal nutrition from the yolk to a requirement for external feeding.

### Newly identified *gck isoform (iso3)* is a major constituent of islet expression

While *gck* isoforms show distinct tissue expression in mammals, differential expression of different isoforms has not been characterized in zebrafish. To directly demonstrate transcription of predicted open reading frames, relative isoform usage in pancreatic cells and embryonic hepatocytes was quantified. The short isoform *gck201* was barely detected across the cell types examined, while the other two identified isoforms were differentially expressed in islet cells and hepatocytes (Fig. 1c). Both *gck202* and *iso3* were expressed in endocrine islet cells. The newly identified *iso3* was expressed in β-cells and α-cells, and was undetectable in δ-cells and hepatocytes. *Gck202* was the predominant isoform in δ-cells, while *gck* in α-cells was 75% *gck202* and 25% *iso3*. In β-cells, *gck202* and *iso3* were expressed at comparable levels (Fig. 1c). In the liver, *gck202* was the prevalent isoform expressed (Fig. 1c). This is similar to what has been shown in mammals, where the isoform with the more proximal first exon (zebrafish *gck202*) is specifically expressed in the liver while the isoform with the distal first exon (zebrafish *gck iso3*) is β-cell specific[31]. Furthermore, amino acid alignment comparing the zebrafish isoforms with the two main GCK isoforms expressed in humans and mice not only showed an 80% similarity, but confirm the alignment of Gck202 to the liver specific isoform 2 while the newly identified Iso3 was more similar to the β-cell specific isoform 1 in Homo sapiens as well as in Mus musculus (Supplementary Fig. 1).

To confirm stage specific dynamics of *gck* expression, we used RNA in situ hybridization. At 4 dpf *gck* expression was observed in the developing liver and in the endocrine pancreatic islet (Fig. 2a). At 5 dpf, *gck* expression was maintained in the islet and increased in the liver (Fig. 2b). At 6 dpf we saw *gck* expression in the islet but no liver specific staining (Fig. 2c), which is in agreement with the analyzed RNA sequencing data showing no or very low expression of *gck* in hepatocytes at 6 dpf (Fig. 1b). At 10 dpf, after larvae were fed a standard diet starting at 6 dpf, *gck* expression was high in the liver, but not detectable in the endocrine islet (Fig. 2d). The lack of islet expression

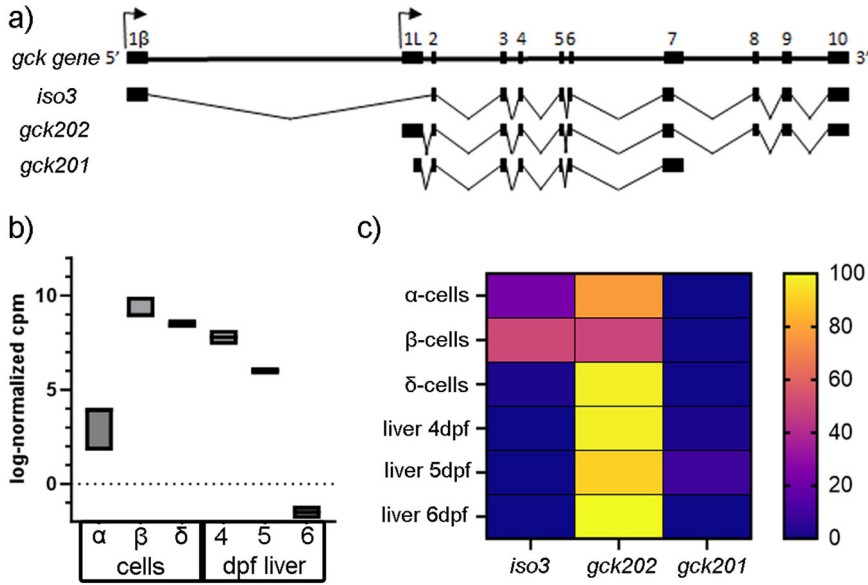

**Fig. 1 | Gck isoforms and their expression in the zebrafish. a** Genomic structure of the zebrafish *gck* gene and the three transcript isoforms. Exons are indicated by black rectangles and introns by lines, curved arrows represent the sites of transcription initiation. **b** Quantification of overall *gck* RNA counts from sorted adult zebrafish pancreatic cells (*N* = 3 biological replicates)[28] and embryonic hepatocytes (*N* = 2 biological replicates for each time point)[29]. Plot shows Min to Max. **c** Heat map showing *gck* isoform expression in sorted zebrafish pancreatic α-, β- and δ-cells and embryonic hepatocytes at 4–6 dpf using the same published datasets[28,29] and the improved zebrafish transcriptome annotation of Lawson et al.[26].

**Fig. 2 | *Gck* shows nutrient dependent expression in the liver and constitutive expression in the endocrine islet at different developmental stages.** Whole mount in situ hybridization at 4 dpf shows *gck* expression in the developing liver as shown in lateral and ventral view (**a** and **a'**) and in the endocrine islet (**a"**). Expression at 5 dpf expands with the development of the liver (**b** and **b'**) and is constitutive in the islet (**b"**). At 6 dpf expression in the liver is almost undetectable (**c** and **c'**), while expression in the islet is visible (**c"**). Expression is prominent in fed 10 dpf larvae in the liver (**d** and **d'**). Expression in the islet in fed 10 dpf larvae is not detectable (**d"**), which is likely due to limited penetration of the RNA-probe. Scale bar: 100 μm and 20 μm.

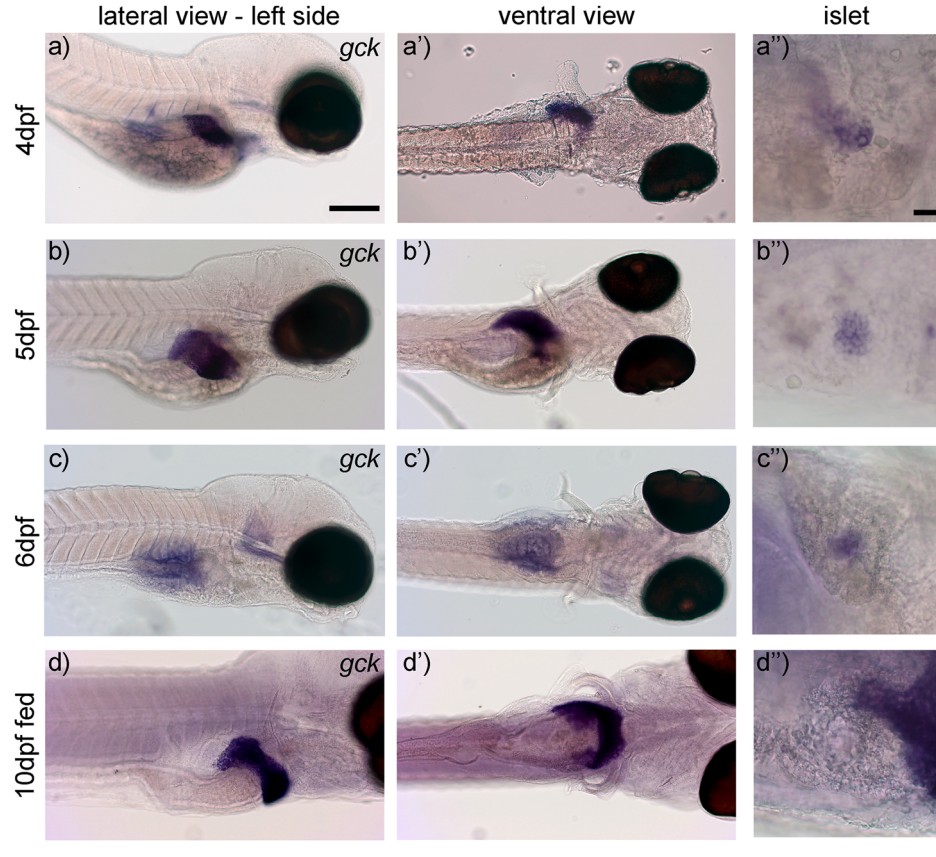

may be a result of low probe penetration into the densely packed endocrine islet.

**Transgenic lines report on *gck* responses to physiological stimuli**
In order to monitor expression of the predominant *gck202* isoform in response to physiological stimuli, we generated a transgenic line $Tg_{BAC}(gck:GFP)$, referred to as *gck:GFP*, in which GFP is expressed downstream of the *gck* ATG (Supplementary Fig. 2). We first examined GFP expression in relation to pancreatic islet cell types. By combining *gck:GFP* with the *ins:dsRed* transgenic line we saw that most β-cells labeled by dsRed also expressed GFP at 5 dpf and in adult zebrafish islets (Fig. 3a and Supplementary Fig. 3a). Antibody staining for Somatostatin showed that all δ-cells were labeled by GFP at 5 dpf and in adult zebrafish (Fig. 3b and Supplementary Fig. 3b), in accordance with published scRNAseq data (Supplementary Fig. 4). Staining for Glucagon revealed very few α-cells expressing GFP at 5 dpf and in adult zebrafish (Fig. 3c and Supplementary Fig. 3c), which is also in agreement with the results from the scRNAseq analysis (Supplementary Fig. 4). In mammals, *GCK* expression in the islet is reported to be constitutive[7]. By quantifying *gck:GFP* expression in the islet of larvae fed a minimal diet (MIN) versus a high fat diet (HFD) from 6 to 10 dpf we confirmed the constitutive expression of *gck* in the endocrine islet (Supplementary Fig. 5a–c). qPCR on isolated islets of 10 dpf larvae and adult fish (> 3 months) confirmed the unchanged expression of *gck* upon increased feeding (Supplementary Fig. 5d, e).

*Gck:GFP* also recapitulated *gck* expression in liver, which commenced around 4 dpf in the developing liver (Fig. 4a) and expanded with tissue growth, maintaining signal intensity at 5 dpf (Fig. 4b). As opposed to the absence of *gck* RNA expression, GFP expression was still visible at 6 dpf in unfed larvae (Fig. 4c), although with a lower intensity compared to 4 and 5 dpf, which can be attributed to the longer stability of GFP. Similar to RNA expression, the transgene expression in the liver depended on nutrition. GFP expression at 10 dpf following HFD feeding was significantly increased compared to larvae fed a minimal diet (Fig. 4d–f). In HFD fed larvae there is

also an increased fluorescent signal in the gut, however this is auto-fluorescence due to ingested food. Moreover, we assessed the effects of different feeding regimens on *gck* expression through the quantification of gck:GFP signal and qPCR and found that HFD increased *gck* expression in a stable and significant manner (Supplementary Fig. 6).

**Diabetic *pdx1* mutants show blunted *gck* responses to nutrients**
β-cell loss and reduced insulin in diabetes is predicted to alter *gck* expression and regulation[8]. To detect tissue-specific changes in *gck* expression in response to perturbed glucose metabolism, we assessed hepatic and islet *gck* expression in diabetic *pdx1* zebrafish mutant larvae and controls before and after feeding a minimal (MIN) or a high fat diet (HFD) (Fig. 5a). Hepatic *gck* expression monitored via *gck:GFP* before commencement of feeding at 6 dpf was 1.5-fold higher in $pdx1^{+/+}$ control animals compared to *pdx1* mutants (Fig. 5b–d). After 5 days of feeding a minimal diet, GFP expression in the liver of both controls and *pdx1* mutants was low to almost undetectable (Fig. 5b', c', e). HFD feeding led to detectable hepatic *gck* expression in control animals (mean Intden of 400) and to highly variable GFP expression in *pdx1* mutants (mean Intden of 300 and a standard deviation of 120) (Fig. 5b", c", e). Generally, *gck:GFP* expression decreased from 6 to 10 dpf, but the decrease was more pronounced when nutrition was minimal as compared to HFD, in control animals as well as in diabetic *pdx1* mutants (Fig. 5f). *Gck* expression levels were confirmed via qPCR (Fig. 5g), highlighting that nutrients modulate *gck* expression in $pdx1^{+/+}$ controls, while HFD in *pdx1* mutants, where insulin signaling is deficient, did not affect *gck* expression. *Gck* expression was significantly increased in controls upon HFD compared to a MIN diet, consistent with physiological effects to augment glucose uptake and storage. Additionally, we also measured Gck enzyme activity in controls and *pdx1* mutants one to two hours after feeding of MIN and HFD diets (Fig. 5h). Gck activity was low in minimally fed controls and *pdx1* mutants. While the activity was up to four-fold increased in $pdx1^{+/+}$ fed HFD, Gck activity in HFD fed *pdx1* mutants remained low, comparable to MIN fed larvae.

**Fig. 3 | Gck:GFP indicates gck expression in islet cells. a** Most β-cells express *gck:GFP* (solid arrows) as observed in 5 dpf double transgenic *gck:GFP;ins:dsRed* larvae. **b** Immunohistochemistry reveals *gck:GFP* expression in all Somatostatin labeled islet cells (solid arrows) at 5 dpf. **c** At 5 dpf, *gck:GFP* expression is rarely observed in Glucagon expressing α-cells (as labeled by anti-gcg antibody, solid arrows). Open arrows highlight hormone producing cells lacking *gck:GFP* staining. Scale bar: 50 μm.

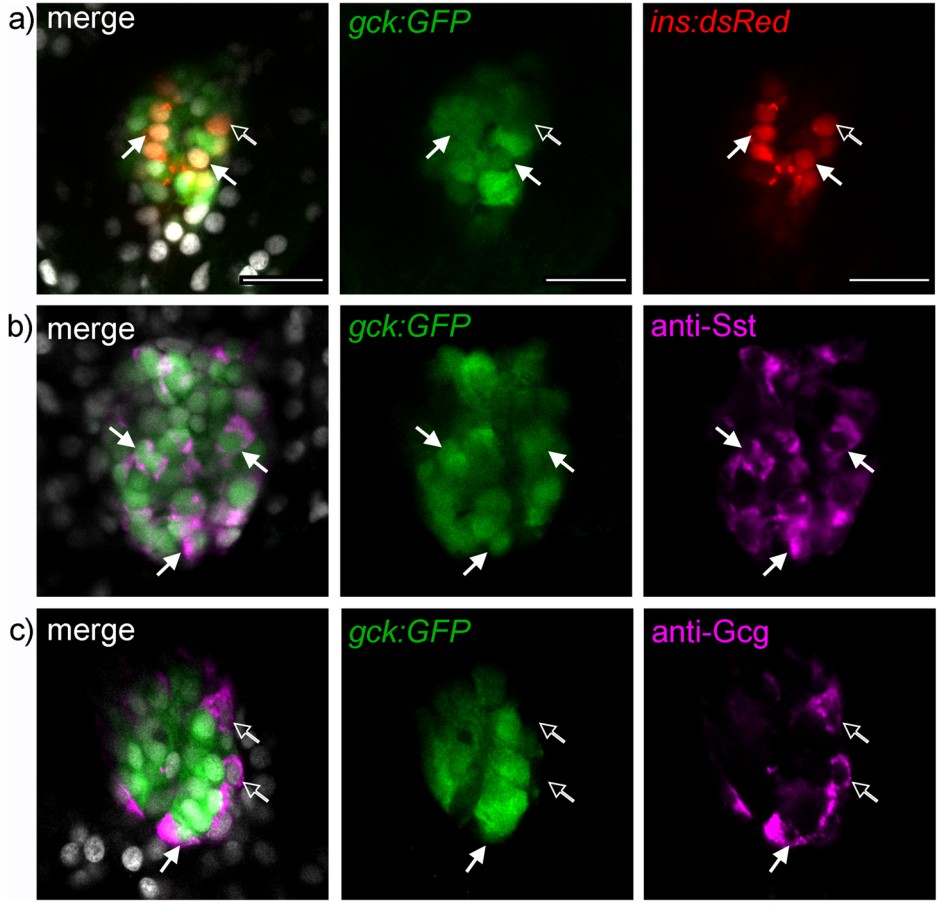

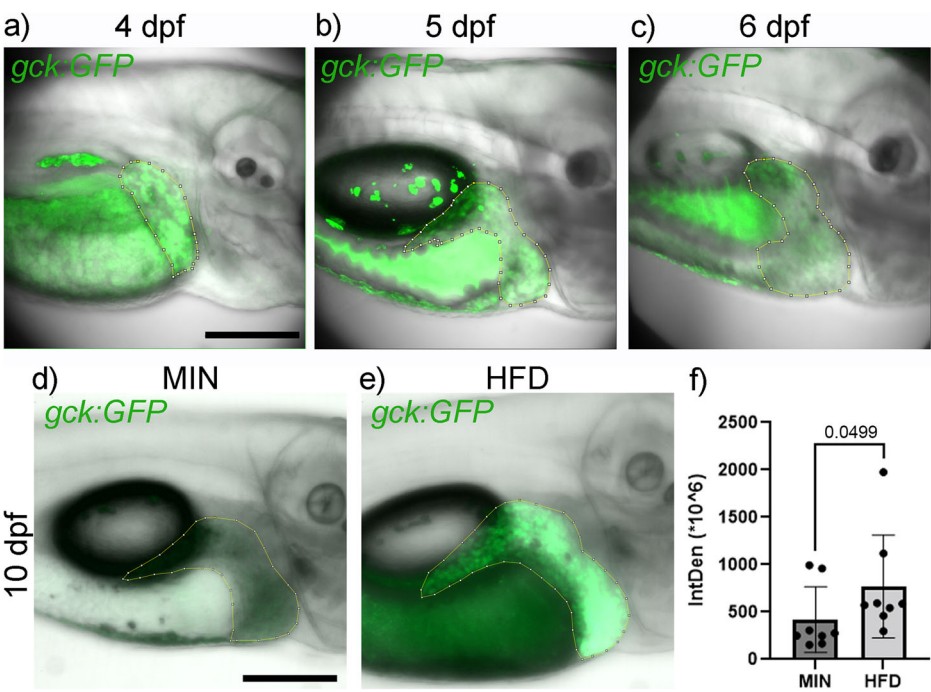

**Fig. 4 | Gck:GFP recapitulates gck expression and nutrient responses in the liver.** Transgenic *gck:GFP* expression in the liver (yellow outline) at 4 dpf (**a**), 5 dpf (**b**) and unfed 6 dpf larvae (**c**). *gck:GFP* expression in the liver is nutrient dependent after depletion of the yolk, showing low expression upon fasting or minimal diet from 6 to 10 dpf (**d**) and significantly elevated expression upon feeding a high fat diet (HFD) from 6 to 10 dpf (**e**). Scale bar: 200 μm. **f** Quantification of GFP signal in the liver of MIN and HFD fed larvae (N = 8). Dot and bar plot showing mean with SD. Statistical significance assessed by Mann–Whitney-test.

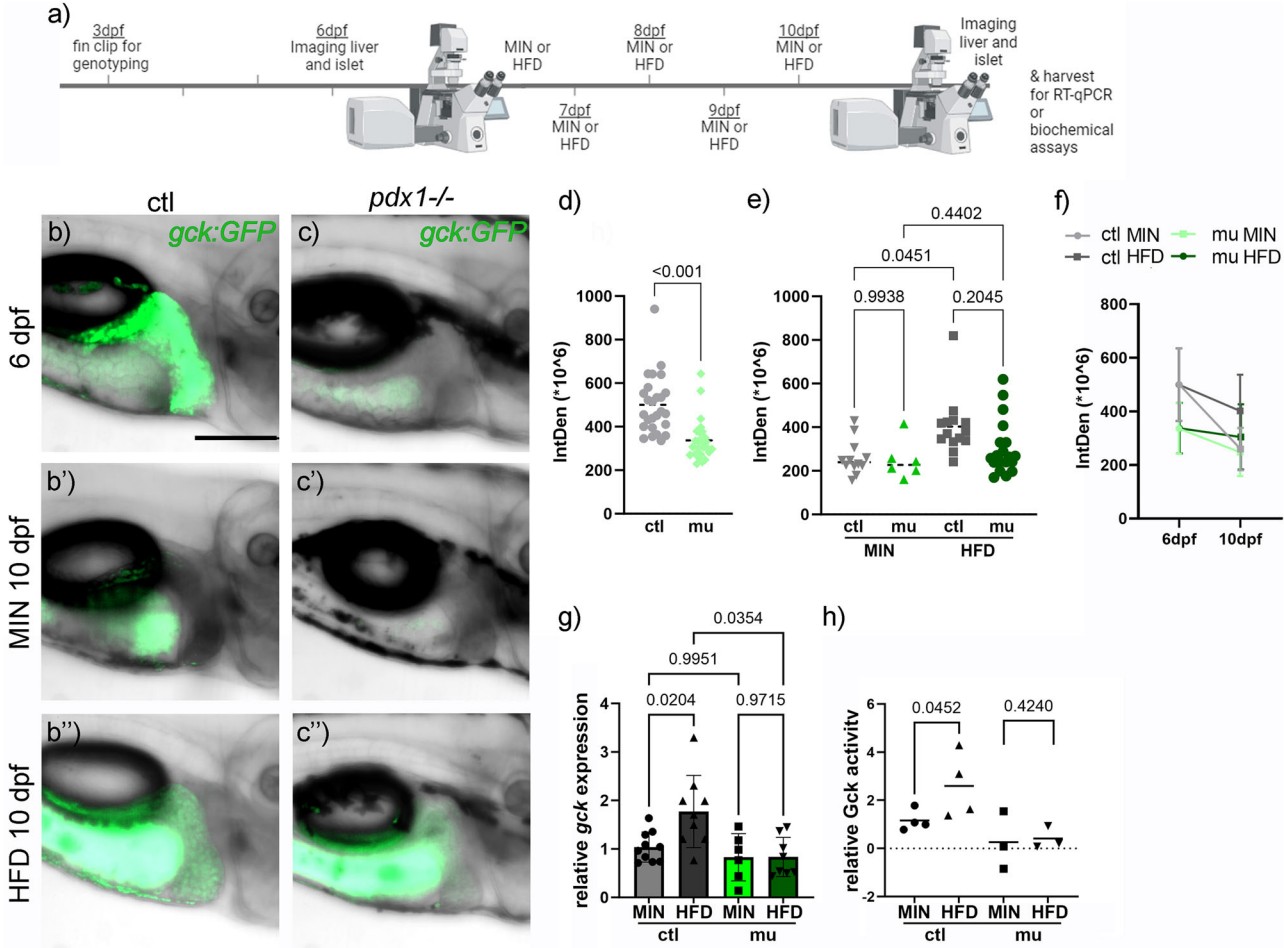

**Fig. 5 | Hepatic *gck:GFP* in diabetic *pdx1⁻/⁻* larvae is unaffected by nutrients.**
**a** Experimental setup and timeline created in https://BioRender.com. Hepatic GFP expression at 6 dpf control and *pdx1⁻/⁻* larvae (**b, c**), and at 10 dpf following 5 days of minimal diet (**b', c'**) or HFD (**b", c"**). Signal in the gut is caused by autofluorescence of ingested food. Scale bar 200 μm. **d** Quantification of GFP signal in the liver of 6 dpf MIN and HFD fed control and *pdx1⁻/⁻* larvae. Dot plots with indicated mean. Statistical significance assessed by t-test. **e** Quantification of GFP signal in the liver of 10 dpf MIN and HFD fed control and *pdx1⁻/⁻* larvae. Dot plots with indicated mean. Statistical significance assessed by Two-way ANOVA, *P*-values of multiple comparisons are indicated. **f** Quantification of GFP signal in the liver of 6 and 10 dpf MIN and HFD fed control and *pdx1⁻/⁻* larvae. Dot plot with indicated mean, SD and lines connecting related groups. **g** *Gck* gene expression levels analyzed via RT-qPCR in MIN and HFD fed control and *pdx1⁻/⁻* larvae (*N* > 3). Dot and bar plot shows mean with SD. Statistical significance assessed by Two-way ANOVA. *P*-values of multiple comparisons are indicated. **h** Measurement of Gck activity in MIN and HFD fed control and *pdx1⁻/⁻* larvae (*N* > 3) shown in a dot plot with indicated mean. Two-way ANOVA indicated a significant effect of the genotype (*P* = 0.0147), but no significant differences in the multiple comparison. Indicated *P*-values are from t-tests.

As *gck:GFP* is also expressed in islet cells, a combination with β-cell labeling using the *ins:dsRed* transgene allowed quantification of overall islet cells and β-cells in control and mutant animals upon feeding a minimal and a HFD. *Gck:GFP* positive islet cells as well as *ins:dsRed* positive cells were reduced by 30–40% in *pdx1* mutants at 6 dpf (Fig. 6a–d). These reduced cell numbers were maintained upon feeding of a minimal or HFD until 10 dpf. Upon a MIN diet, *ins:dsRed* positive cells were reduced by around 60% in *pdx1* mutants, while *gck:GFP* positive cells were reduced by around 50% in mutants compared to *pdx1⁺/⁺* controls (Fig. 6a',b',e, f). Upon HFD feeding, *ins:dsRed* positive and *gck:GFP* positive cells were reduced by around 40% in *pdx1* mutants (Fig. 6a",b",e, f). Consistent with these findings, *insulin* RNA expression increased by almost fivefold in control animals upon feeding HFD compared to a MIN diet, while *pdx1* mutant animals were not able to respond in a similar manner to an enhanced nutrient supply due to the reduced number of β-cells (Fig. 5g).

Overall, *insulin* expression and β-cell number were reduced in *pdx1* mutants and did not increase upon nutrient stimulation. Similarly, the expression of hepatic *gck* and Gck activity did not respond to nutritional stimuli in diabetic mutants.

## Gck activation reversed hyperglycemia in *pdx1* mutants

GCK inhibition has been proposed as a potential therapeutic strategy to enhance β-cell rest and recovery, while the use of GCK activators (GKA) to treat diabetes has been plagued by issues of efficacy and adverse effects[11,14,15]. We used our *pdx1* mutant zebrafish to test Gck modulation with cellular readouts in a whole organism diabetes model.

Based on the above findings, we hypothesized that treatment with a GKA may boost hepatic glucose uptake and normalize glucose levels in diabetic animals, while treatment with a Gck inhibitor might increase glucose levels further. Diabetic *pdx1* mutant larvae and controls were treated with 2 μM Dorzagliatin, 500 μM D-Mannoheptulose or DMSO for 5 days with feeding (Fig. 7a). Glucose measurement in 10 dpf larvae revealed significantly elevated glucose levels in *pdx1* mutant larvae (almost four-fold), which were ameliorated by treatment with Dorzagliatin (Fig. 7b). Glucose levels in larvae treated with D-Mannoheptulose were further elevated as compared to DMSO treated *pdx1* mutants, but this was not significant (Fig. 7b). To confirm that the effects on glucose levels were due to modulation of Gck activity, we measured Gck activity in *pdx1* mutant larvae treated with Dorzagliatin and D-Mannoheptulose (Fig. 7c). Dorzagliatin

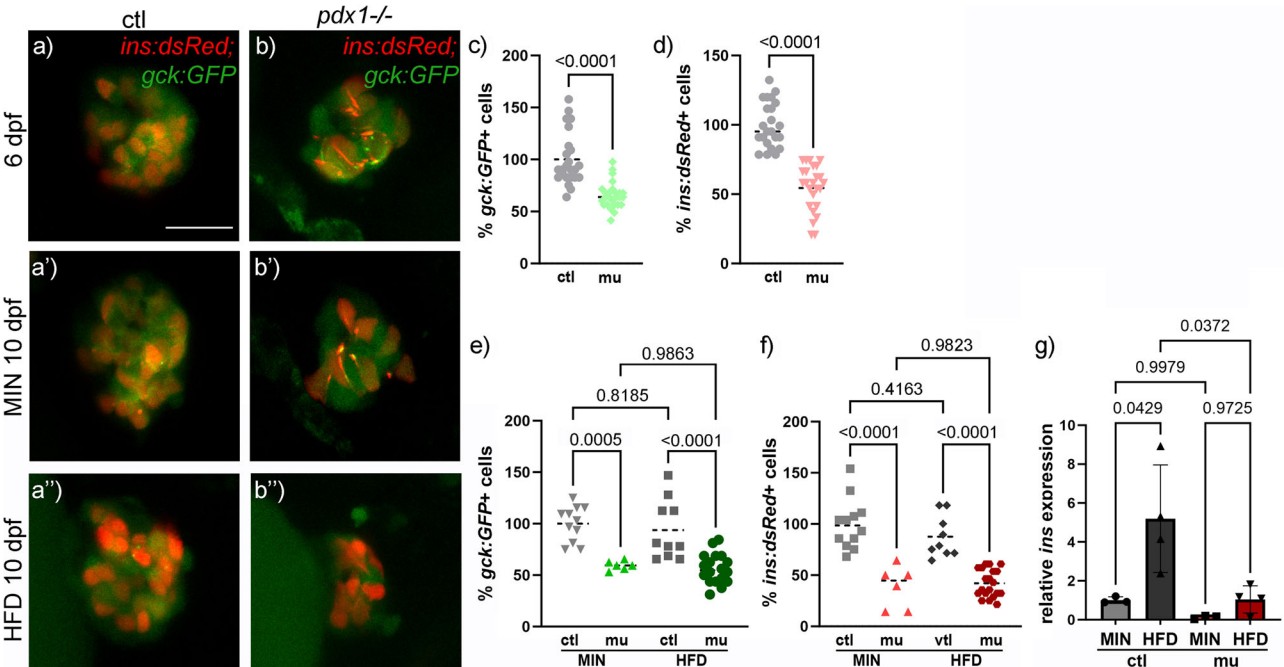

**Fig. 6 | Gck:GFP expression in the islet in pdx1⁻/⁻ diabetic larvae is reduced.**
**a, b** gck:GFP and ins:dsRed labeled cells in the islets of 6 and 10 dpf control and pdx1⁻/⁻ after MIN and HFD diet. Scale bar 50 μm. **c, d** Relative quantification of gck:GFP expressing cells and ins:dsRed labeled β-cells in the islet of pdx1⁻/⁻ compared to controls at 6 dpf. Dot plots with indicated mean. Statistical significance assessed by t-test. **e, f** Relative quantification of gck:GFP expressing cells and ins:dsRed labeled β-cells in the islet of pdx1⁻/⁻ compared to controls at 10 dpf. Dot plots with indicated mean. Statistical significance assessed by Two-way ANOVA, P-values of multiple comparisons are indicated. **g** Insulin gene expression levels analyzed via RT-qPCR under MIN and HFD feeding (N > 3). Dot and bar plot shows mean with SD. Statistical significance assessed by Two-way ANOVA. P-values of multiple comparisons are indicated.

treatment significantly increased Gck activity around 1.4 fold compared to pdx1 mutants treated with DMSO. D-Mannoheptulose treatment appeared to reduce Gck activity, but not significantly. To gain further insight into the effects of Dorzagliatin, we assessed glucose uptake in the liver using the fluorescent glucose analog 2-NBDG at the end of the treatment. A significant increase of 2-NBDG was observed in the liver of Dorzagliatin treated pdx1 mutants (Supplementary Fig. 7a–c). This was accompanied by a significant increase in expression of the insulin-independent glucose transporter glut2 (Supplementary Fig. 7d). Further gene expression analysis revealed a trend towards upregulation of genes related to glycolysis (hexokinase 1 (hk1) and pyruvate kinase L/R (pklr)), and of insulin, but no increased expression of gck itself in treated pdx1 mutants (Supplementary Fig. 7d).

It has been previously reported that treatment with GKAs leads to β-cell stress and steatosis in the liver[32]. To examine this, we assessed the co-expression of hepatic gck:GFP and an oxidative stress reporter line Tg(EpRE:RFP)[33]. Hepatic GFP expression was reduced in pdx1 mutants irrespective of treatment with DMSO, Dorzagliatin or D-Mannoheptulose (Fig. 7d, f). RFP expression, as a readout for oxidative stress in the liver of treated pdx1 mutants, was not significantly changed compared to DMSO-treated pdx1 mutants and pdx1⁺/⁺ controls (Fig. 7e, f).

Using the same reporter line, we also looked for oxidative stress responses in islets of controls and pdx1 mutants treated with the GKA Dorzagliatin (Fig. 8a). In controls we did not detect any signs of oxidative stress in or surrounding the primary endocrine islet, irrespective of treatment. Interestingly, DMSO and Dorzagliatin treated pdx1 mutants showed signs of oxidative stress, notably not in the primary endocrine islet but in proximity to the islet. Additionally, we looked for changes in β-cell numbers in controls and pdx1 mutants expressing ins:nuc-RFP and did not observe negative effects of Dorzagliatin treatment on β-cell numbers (Fig. 8b, c).

Altogether, Gck activation by Dorzagliatin ameliorated elevated glucose levels in pdx1 mutant larvae without inducing oxidative stress in the liver or the endocrine islet and without negative effects on β-cell number.

## Discussion

Our experiments showed highly conserved expression and regulation of Gck, an enzyme playing a critical role in glucose homeostasis, in the vertebrate model organism zebrafish. They further elucidated gck regulation under pathological conditions in diabetic pdx1 mutant zebrafish, where β-cell number and insulin expression are reduced. In diabetic zebrafish, hepatic gck expression and activity were low and this did not increase with a nutrient load, as occurs in normoglycemic controls. As a demonstration of Gck treatment responses in a whole-organism model, we showed that Gck activity can be modulated by the GKA Dorzagliatin to ameliorate hyperglycemia in pdx1 mutants, which occured without detectable activation of oxidative stress responses in liver or endocrine β-cells. By contrast, Gck inhibition in a diabetic organism did not show beneficial effects on glycemia.

Previously, only a glucose-responsive hexokinase expressed in liver was identified in the zebrafish[24]. By using enhanced annotations, we were able to identify three gck isoforms, which show genomic organization and mRNA expression patterns similar to mammalian isoforms[31]. Our results confirm and expand on earlier studies in zebrafish[24,30,34], now demonstrating that liver expression of gck behaves similarly to the mammalian liver GCK in response to fasting and nutrients at larval stages, when drug treatment can be readily applied. In contrast to short-term feeding responses previously reported, we assessed the effects of feeding over days under normoglycemic and hyperglycemic conditions. By using pdx1 mutant zebrafish, which have reduced insulin signaling[23], we show evidence for regulation of liver gck expression mainly by insulin. The gck:GFP transgene recapitulated gck expression as predicted in our analysis of RNA sequencing data and ISH, both in the liver and pancreatic islet, facilitating analyses through live-imaging approaches in transparent zebrafish larvae. Consistent with findings in mammals, where GCK expression is not limited to β-cells, zebrafish gck was also expressed in islet α- and δ-cells, where we predict it similarly regulates glucose sensing and hormone secretion[35,36]. We observed heterogeneous gck:GFP expression in β-cells in larval and adult zebrafish. This was confirmed in published

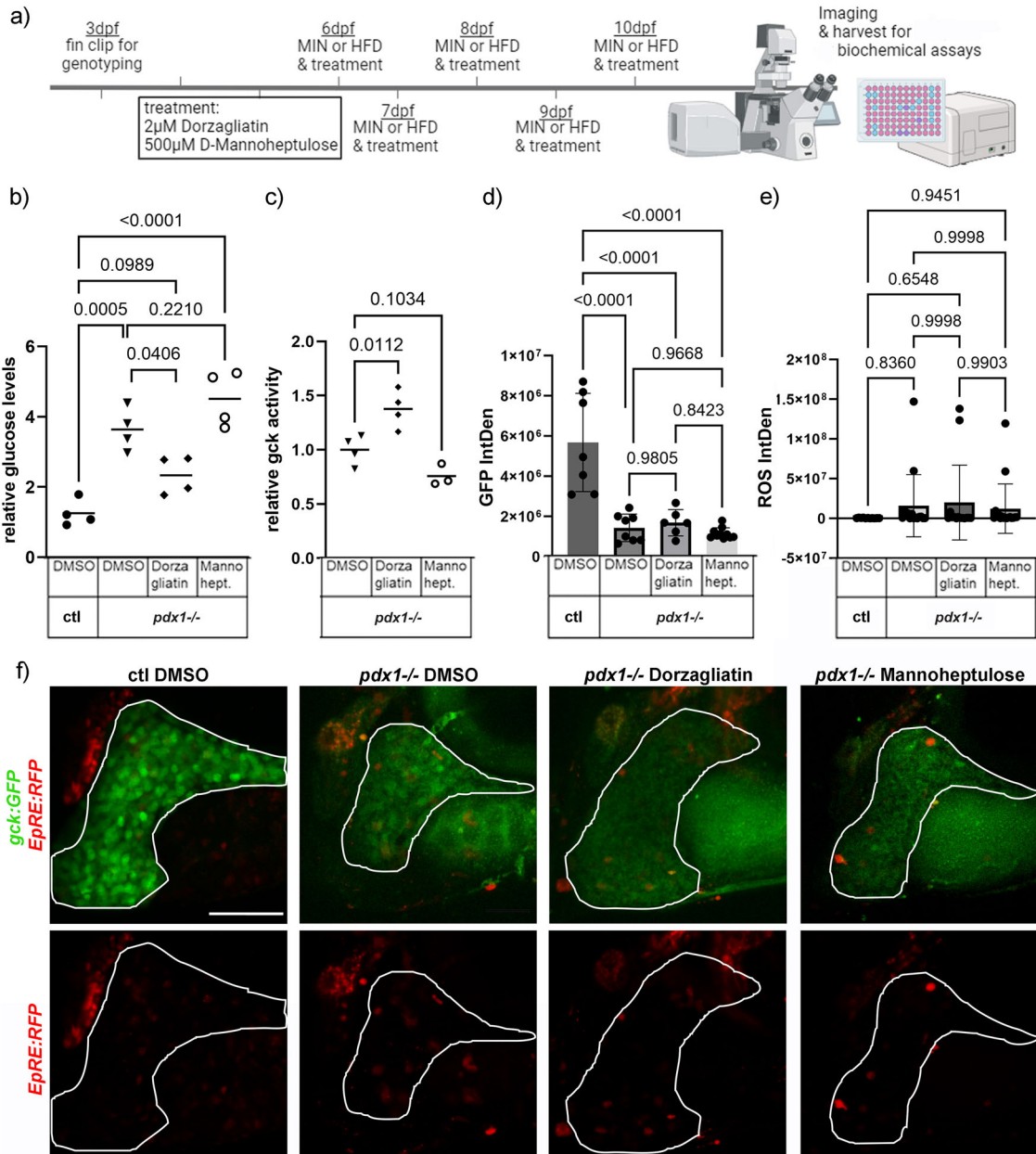

**Fig. 7 | Activation of Gck ameliorates hyperglycemia. a** Experimental setup and timeline created in https://BioRender.com. **b** Glucose measurements of *pdx1*[-/-] after 5 days treatment with either 2 µM Dorzagliatin or 500 µM D-Mannoheptulose reveal a glucose lowering effect of Dorzagliatin (N = 4) as shown in a dot plot with indicated mean. Statistical significance assessed by One-way ANOVA. *P*-values of multiple comparisons are indicated. **c** Measurement of Gck activity in *pdx1*[-/-] following treatments as indicated (N > 3) as shown in a dot plot with indicated mean. Statistical significance assessed by One-way ANOVA. *P*-values of multiple

comparisons are indicated. **d** *gck:GFP* expression following Dorzagliatin or D-Mannoheptulose treatment in *pdx1*[-/-] larvae. Stress-induced fluorescence signal quantification (**e**) and representative images (**f**) showing *gck:GFP* and oxidative stress as assessed by *EpRE:RFP* expression in the liver of control DMSO-treated and *pdx1*[-/-] treated with DMSO, Dorzagliatin or D-Mannoheptulose. (Liver is outlined in white). Dot and bar plot showing mean with SD. Statistical significance assessed by One-way ANOVA. *P*-values of multiple comparisons are indicated. Scale bar: 100 µm.

single cell RNA sequencing data of zebrafish pancreatic islet from Singh et al.[37] and single cell RNA sequencing data of human pancreas from Baron et al.[38], showing that *GCK* expression in β-cells is diverse.

In diabetic *pdx1* mutants, we could determine effects of low insulin and elevated glucose on *gck* expression and activity. *Gck* was reduced in islets and liver of diabetic mutants compared to controls, and liver expression, as well as its activity, did not change with feeding in *pdx1* mutants, consistent with insulin-dependent regulation. This is furthermore in accordance with decreased GCK activity reported in patients with type 2 diabetes[39].

Since 2003, numerous GKAs have been developed, but many have been discontinued because of efficacy and safety issues, including dyslipidemia,

hypoglycemia, and unsustained glycemic control[40]. While ongoing clinical trials of the dual-acting GKA, Dorzagliatin, and the hepato-selective GKA, TTP399, show promising efficacy and no concerning side effects so far[13], inhibition of GCK is considered to be a counter-intuitive but possibly beneficial alternative approach[11,14,15]. The concept of GCK inhibition as a potential treatment for diabetes is based on the correction of islet over-activity early during diabetes by modifying the cellular metabolism of β-cells and decreasing their secretory stress[14,15]. Support for this theory comes from studies of heterozygous inactivation of GCK in human and mouse, which showed that reduced GCK activity produces a small rise in fasting blood glucose but neither progressive decline of β-cell function nor increased risk

**Fig. 8 | GKA Dorzagliatin does not induce oxidative stress in β-cells. a** *gck:GFP* and *EpRE:RFP* expression in the endocrine islet (highlighted in white) of controls and *pdx1*[-/-] treated with DMSO and GKA Dorzagliatin. Scale bar: 20 μm. **b** β-cells labeled with H2B-RFP in control and *pdx1*[-/-] larvae treated with DMSO and Dorzagliatin. **c** The absolute β-cell numbers are unchanged upon treatment with Dorzagliatin in controls and *pdx1*[-/-] (*N* > 3) as shown in a dot plot with indicated mean Statistical significance assessed by One-way ANOVA (<0.0001), *P*-values of multiple comparisons are indicated.

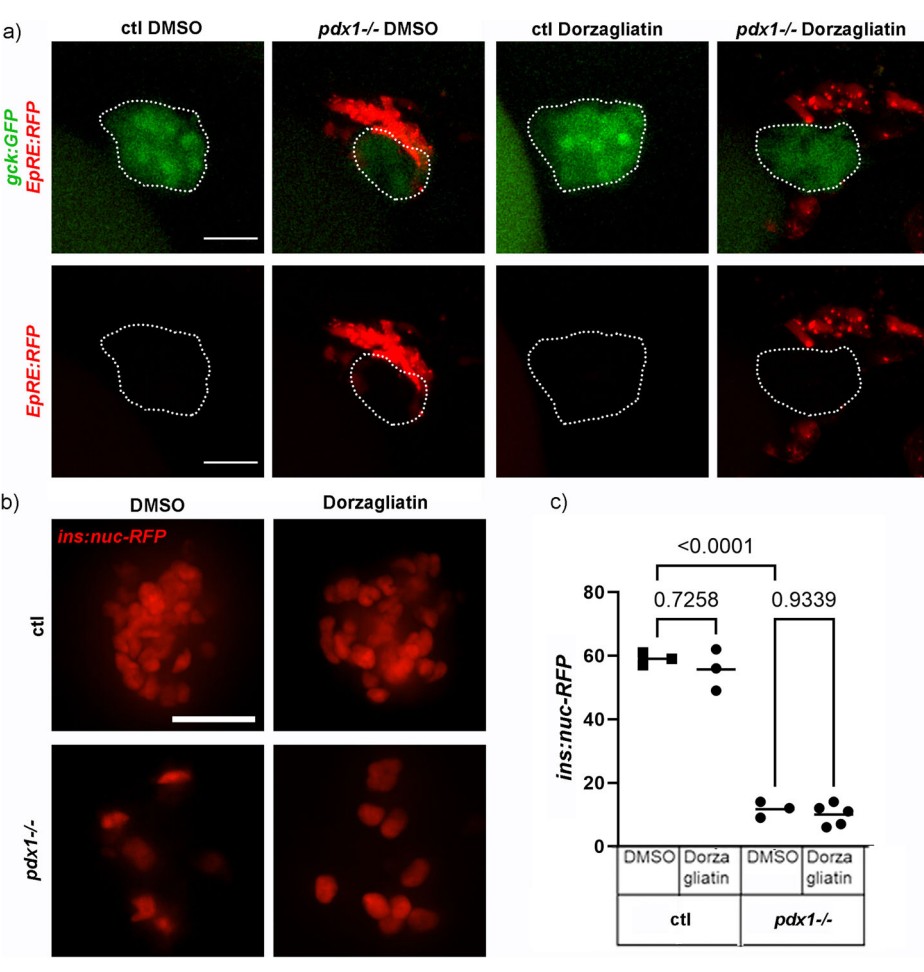

of diabetic complications[14,15]. More evidence comes from the in vitro application of D-Mannoheptulose, a competitive inhibitor of hexokinases, which reversed detrimental effects of hyperglycemia on β-cells and islets[11,14,15]. However, the potential clinical value of treatment with D-Mannoheptulose is limited since it is active in the millimolar range, and inhibition of hepatic GCK is predicted to impair glucose storage and lead to a rise in blood glucose levels[15], which we observed as a trend in our diabetic zebrafish model. More robust effects on glycemia and Gck activity may be seen with Mannoheptulose treatments at doses higher than the 500 μM used in this study. Still, outcomes may also be obscured by inhibition of other hexokinases by D-Mannoheptulose. This issue can be further clarified once specific GCK inhibitors become available. Hence, this work highlights that GKIs must act selectively and potently on overactive β-cells to be of potential benefit in diabetes.

We examined metabolic effects of Gck modification in the *pdx1* mutant zebrafish, which resembles human neonatal diabetes in its early onset, but features a population of β-cells with reduced viability and high vulnerability to stress, as in Type 2 diabetes[23]. The hyperglycemia modulating effects of Gck activation by Dorzagliatin could be confirmed in our zebrafish diabetes model. We hypothesize that the observed decrease in glucose levels was primarily due to effects on the liver, given the reduced numbers of β-cells in the *pdx1* mutant larvae, and the large population of hepatocytes. The observed increase in 2-NBDG uptake in the liver, *glut2* upregulation, and a trend towards elevated expression of genes related to glycolysis strengthen this hypothesis. Nevertheless, Dorzagliatin treatment might also affect *insulin* expression in β-cells in the *pdx1* mutants, indicated via a trend towards elevated *insulin* expression. By making use of a biosensor to assess oxidative stress, we were able to rule out that Dorzagliatin caused liver or β-cell stress within the duration of treatment. Interestingly, we detected a stress response in cells near the islet in *pdx1* mutants, which was

absent in controls. Although we did not pinpoint the affected cell type, such responses are consistent with a role for oxidative stress in diabetic complications, particularly affecting vascular endothelium[41].

With this study we define *gck* isoforms in zebrafish and demonstrate effects of Gck modulation in a whole-organism diabetes model. We acknowledge that further investigations can provide additional important insights. Although nutrients and especially carbohydrates have been shown to mainly control *gck* expression in zebrafish[24,42], our data from the diabetic *pdx1* mutants also point towards insulin as a regulator of *gck* expression. It is also worth mentioning that there are differences in macro- and micro-nutrient requirements in the fish diet compared to human dietary requirements, as the zebrafish is less dependent on dietary carbohydrates[43,44]. In mammals, transcription and protein levels, as well as posttranslational modifications and compartmentalization of GCK, are regulated differently in β-cells and in the liver[45]. Therefore, transgenes or other assays to monitor Gck activity and localization in zebrafish in vivo could give additional insights into tissue specific regulation and further our understanding of whole-organism Gck modulation. Furthermore, to completely rule out negative effects on islet and liver physiology from chronically applied treatments, Gck modulation should be monitored over longer time periods.

Here we show the value of zebrafish diabetes models, where the interplay of metabolic organs determines glucose levels in ways similar to humans, for evaluating potential treatments for a complex disease such as diabetes. Zebrafish larvae are amenable to pharmacologic interventions and permit prompt assessment of efficacy and adverse effects by cellular read-outs, which we have demonstrated for the GKI D-Mannoheptulose and GKA Dorzagliatin. Using the baseline conditions and assays established here, novel GCK modulating agents and the effects of short-term and long-term treatments can be evaluated, which will help to clarify fruitful directions for exploring GCK-targeting pharmaceutical agents.

## Research design and methods

### Animal husbandry

Male and female zebrafish (Danio rerio) were maintained according to standard protocols. Lines were kept in a mixed Tuebingen and Mitfa[b692/b692]/ednrb1[b140/b140] background (a gift from Wolfgang Driever, University of Freiburg, Germany). Pdx1 mutants and pdx1[+/+] controls were generated from incrosses of heterozygous parents and genotyped as previously described[23]. In short, genomic DNA prepared from embryo and adult fin clips was genotyped by PCR (pdx1_forw: CCCCAACGAAGACTA-CAGCC, pdx1_rev: ATGGCCTGCAATCAGGAGTTA) followed by restriction digest with DraI. Studies used non-transgenics and the previously generated lines ins:dsRed[m1018Tg](in short: ins:dsRed)[46] and ins:loxP-NLS-mCherry-loxP-DTA[bms525Tg] (in short: ins:nuc-RFP)[47]. All procedures were approved by the Austrian Bundesministerium für Wissenchaft und Forschung (GZ BMWFW-66.008/0018-WF/V/3b/2017, GZ. 2020-0.282.289). We have complied with all relevant ethical regulations for animal use.

### Generation of transgenic lines

The gck BAC construct Tg_BAC(gck:GFP) was generated using the BAC clone CH73-270I8 containing the gck locus and primers listed in Supplementary Table 1. All recombineering steps were performed as described by Bussmann and Schulte-Merker[48], with the modifications as described in Helker et al.[49]. Tg(EpRE:RFP) was generated by injection of the Tol2-3EpRE-hsp70-mCherry-polyA-Tol2 plasmid[33] (generously provided by Dr. Tetsuhiro Kudoh and Dr. Aya Takesono), according to standard protocols[50].

### Bioinformatic analysis

The Lawson Lab Transcriptome Annotation, Genomic Annotation V4.3.2.gtf (https://www.umassmed.edu/lawson-lab/reagents/zebrafish-transcriptome/) was loaded into the integrated genome viewer (IGV.org) and compared against the Ensembl annotation of gck. Modifications were used for downstream RNAseq quantification of gck expression levels. To determine tissue and cell-type variation in isoform expression we used published datasets of sorted adult zebrafish pancreatic cells[28] (available at: https://www.ebi.ac.uk/ena/browser/view/PRJEB10140) and embryonic hepatocytes[29] (available at: https://www.ebi.ac.uk/ena/browser/view/PRJNA849822), both using fish from the AB strain. Raw RNASeq reads[28,29] were retrieved and aligned against the zebrafish genome (GRCz11)[51] using STAR[52]. Modified transcript annotations from v4.3.2[26] were used for gene and isoform level quantification using RSEM[53]. RSEM transcript quantifications were imported into R and summarized to gene level using tximport[54]. Library size normalized counts were obtained using edgeR[55]. Relative isoform usage for each sample was quantified directly via RSEM. A more detailed description of the bioinformatic analysis and data used to generate the graphs can be found in the Supplementary Methods.

### In situ hybridization (ISH)

Full length gck202 was amplified from cDNA using the primers listed in Supplementary Table 1. ISH was performed with a fragmented digoxigenin-labeled antisense RNA probe (DIG RNA Labeling Mix, Roche) and anti-digoxigenin-AP antibody (1:4000, Roche) using previously published protocols[56] in embryos at 4, 5 and 6 dpf. Larvae at 10 dpf were stained according to the protocol published by Vauti et al.[57].

### Immunohistochemistry

Larvae at 5dpf were fixed in 4% PFA overnight at 4 °C, washed three times in PBST and deskinned to facilitate penetration. Then, the samples were permeabilized and blocked with a solution containing 0.1% TritonX-100, 1% BSA, 5% sheep serum. Primary antibodies staining for somatostatin (Dako, A0566), glucagon (Sigma-Aldrich, G2654) and GFP (Aves Lab, GFP-1010) were applied using 1:200 dilutions and secondary antibodies anti-Chicken, Alexa Fluor Plus 488 (Invitrogen, A32931), anti-Mouse, Alexa Fluor 546 (Invitrogen, A-11030) and anti-Rabbit, Alexa Fluor 568 (Molecular Probes, A-11011) were applied using 1:1000 dilutions. Imaging was performed on a Zeiss LSM700.

### RT-qPCR

Total RNA samples were prepared using TRI Reagent (MRC, TR118). cDNA was prepared using the Maxima First Strand cDNA Synthesis Kit (Thermo Fisher Scientific, K1641). HOT FirePol EvaGreen qPCR Mix (Solis BioDyne, 08-31-00020) was used for qPCR reactions in a CFX Connect Real-Time System (Bio-Rad). For RNA extraction three to five larvae were pooled. At least three biological samples for each genotype and treatment were measured in two technical replicates. Primers are listed in Supplementary Table 1.

### Feeding protocols

Minimal feeding (MIN) was performed once per day by adding about 0.75 mg/larva Zebrafeed <100 μm (Sparos, composition: 63.4% protein, 11.3% ash, 13.4% fat, 3.6% carbohydrates). High fat diet (HFD) feeding was performed twice per day by adding about 0.75 mg/larva Zebrafeed <100 μm and 2 μl of a 2% egg yolk (Eierei, composition: 55.8% fat, 3.6% carbohydrates, 34.3% protein) solution per larva.

### In vivo Image acquisition

6 and 10 dpf larvae were anaesthesized with tricaine and embedded in 1.2% low melt agarose. The endocrine pancreas was imaged on a Zeiss LSM700, while the liver was imaged on a Zeiss Axio Observer.Z1 with a Yokogawa CSU-X1 spinning disk, consecutively.

### Islet cell number quantification

Quantification of ins:dsRed and gck:GFP positive cells in the pancreas was performed with Imaris (Bitplane) using the Spot Detection function with a spot diameter of 4 μm in 3D visualizations of confocal z-stacks spanning the region.

### Quantification of fluorescence signal from *gck:GFP* and *EpRE:RFP*

Gck:GFP and EpRE:RFP signal in the liver was quantified in ImageJ by making a Z projection using the 'Sum Slices' option. The liver was manually outlined and the integrated density (IntDen) of GFP or RFP signal in the selected region of interest was measured.

### Drug treatments

Dorzagliatin (MedChemExpress, HY-109030) and D-Mannoheptulose (Cayman Chemical, 16546) were dissolved in DMSO to prepare stock solution. Compounds were administered in egg water at final concentrations of 2 μM Dorzagliatin and 500 μM D-Mannoheptulose. Water was refreshed every day and new compound was added.

### Glucose measurement

Glucose measurements were performed using the Biovision Glucose Assay Kit (ab65333, Abcam), according to manufacturer's instructions with minor adjustments[23]. Three to five 10 dpf larvae were pooled for each sample. Four biological replicates for each genotype and treatment were measured in two technical replicates.

### Glucokinase activity measurement

Gck activity measurement was performed using a fluorometric assay kit (ab273303, Abcam), according to manufacturer's instructions. Tissue extracts were prepared by homogenization of pooled 10 dpf larvae (4 to 11) with glass beads in 250 μl cold 2.5 mM DTT in gck assay buffer. Extracts were spun for 10 min, 12,000 rpm at 4 °C to remove debris and the supernatant was further diluted 1:10 in gck assay buffer. Standard curves were generated and positive controls were measured with every assay. Sample background was assessed if necessary. The assay was performed immediately. Measurement of fluorescence was performed in kinetic mode for 30 min on an EnSpire® Multimode Plate Reader (PerkinElmer). Gck activity was normalized to protein concentration of the extract, number of pooled larvae and is reported relative to the average control value. At least three biological replicates for each genotype and treatment were measured in two technical replicates.

## Statistics and reproducibility

To guarantee reproducibility and avoid bias, the larvae in an experiment came from pooled batches of eggs and experiments were repeated at least two times. Sample sizes, number of replicates and definition of replicates were as following: For RNA extraction three to five larvae were pooled per biological sample. At least three biological samples for each genotype and treatment were measured in two technical replicates. For glucose measurements three to five larvae were pooled for each biological sample. Four biological replicates for each genotype and treatment were measured in two technical replicates. For glucokinase activity measurements tissue extracts were prepared by homogenization of pooled larvae (4–11). At least three biological replicates for each genotype and treatment were measured in two technical replicates. For graphs analyzing images, scatter dot plots were used for representation to confer sample size. All data were analyzed and graphs were generated using Graphpad Prism10.2.0. Statistical significance was determined using t-test, one-way ANOVA, two-way ANOVA and appropriate post-hoc tests as stated in the corresponding figure legends. Results were considered statistically significant if their corresponding $p$-value was less than 0.05. Exact $p$-values are indicated in the graphs.

## Reporting summary

Further information on research design is available in the Nature Portfolio Reporting Summary linked to this article.

## Data availability

All source data underlying the graphs presented in the Figures and Supplementary Figs. are uploaded as Supplementary Data. All other data are available from the corresponding author upon reasonable request.

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

## Acknowledgements

The authors thank Dzenana Tufegdzic for expert zebrafish care, Sonja Toechterle and Thomas Waldner for excellent technical assistance. We thank Dr. Tetsuhiro Kudoh and Dr. Aya Takesono (University of Exeter) for providing the Tol2-3EpRE-hsp70-mCherry-polyA-Tol2 plasmid. Schematics of experimental setup and timelines were created with BioRender.com. This work received funding from the Tiroler Wissenschaftsfond (N.S), the Austrian Science Fund (FWF, grant DOI: 10.55776/P30038, R.A.K., N.S.) and the European Union Horizon 2020 Research and Innovation Programme, under the FETOPEN grant agreement No. 899612 (SWIMMOT Project, R.A.K., N.S.). For open access purposes, the author has applied a CC BY public copyright license to any author accepted manuscript version arising from this submission.

## Author contributions

Conceptualization: N.S. and R.A.K.; Methodology: N.S., E.M. and C.H.; Validation: N.S.; Formal analysis: N.S. and D.R.; Investigation: N.S., S.T, E.M. and I.B.; Resources: D.Y.R.S., D.M. and R.A.K; Writing and Editing: N.S. and R.A.K.; Visualization: N.S.; Supervision: N.S., D.M. and R.A.K.; Funding acquisition: N.S. and R.A.K.; N.S. is the guarantor of this work and, as such, had full access to all the data in the study and takes responsibility for the integrity of the data and the accuracy of the data analysis.

## Competing interests

The authors declare no competing interests.
