## [Peer review file · Communications Biology]

Conserved glucokinase regulation in zebrafish confirms therapeutic utility for pharmacologic modulation in diabetes

Corresponding Author: Dr Nicole Schmitner

Version 0:

Reviewer comments:

Reviewer #1

(Remarks to the Author)

This is an exciting paper, developing a novel zebrafish model to help investigate therapeutic interventions for diabetes and other metabolic disorders. The data are well presented and, for the most part, very clearly presented. There are some minor edits that should be made to help with the clarity of the models that were used in the studies.

Minor edits:

Fig 1- It is a little unclear where the data come from for Fig. 1. I am assuming the data presented were from archived data from the paper cited and analyzed by the authors to generate the figure. It would be helpful if this was more clearly stated in the text. Also, differences in zebrafish strains used for these previous studies should be noted.

line 115: "liver anlage"- I am not sure what this refers to. I am not an anatomist, but I could not find any ref to this elsewhere.

line 121 "We attribute the absence of islet expression to probe penetration problems". Although this is a logical assumption, without any supporting data, you really cannot attribute it to anything. I would change the wording to something like "lack of islet expression may be a result of low probe penetration".

All zebrafish models

1. "wild-type" does this mean you are using an outbred set of fish? If not you need to say what wild-type line you were using.
2. For each experiment the transgenics and their appropriate controls need to be discussed. For example in Fig 5/ 6 it was not clear whether the animals were the *pdx*^{-/-}; *gck*:GFP or if they were all *pdx*^{-/-}; *gck*:GFP;*ins*:*dsRED*. For clarity, it would be good to clearly identify the genetic lines for each figure.

Reviewer #2

(Remarks to the Author)

This manuscript by Schmitner et al., address an important biological question: glucokinase regulation in the zebrafish. Although this manuscript present novel an interesting data, it also has some flaws that would need to be rectified before publication in Communications Biology.

Major points:

- 1- Is *gck* expression in the islet always constitutive/? Even in adults
- 2- The first sentence in the introduction (L39-40) needs a reference
- 3- In the introduction the authors should present the hexokinase family mention that CSK is hexokinase 4 and introduce the differences between HK1,2 and HK4 as they later use mannoheptulose
- 4- L64 define T2D
- 5- L66 replace beta-cells but beta cell number
- 6- L68: *gck* isoforms has not been well described... but what has been described so far?
- 7- Fig. 1b and c: how was this figure generated? How were the RNA counted? 1C how did the authors discriminate between the three *gck* isoforms?
- 8- There is no description of the ScRNAseq experiment used to produce the data presented in Fig. 1b
- 9- What is the carbohydrate source in the HFD experiment?
- 10- Figure 4 is confusing... Were these larvae fed at all? Did these larvae follow the same feeding protocol (feeding start on day 6) as in figure 3? If so were the 6 dpf larvae fed?
- 11- The main difference in the MIN vs HFD diet is the amount of dry food received by the larvae (twice as much in the HFD protocol). As egg yolk is not rich in carbohydrates the difference observed between Figure d and e can be solely attributed to the amount of carbs fed to the larvae in dry food and be totally independent form an HFD (excess fat) regime. The author

- should repeat this experiment with excess food, not HFD diet to confirm that the increase of GFP localization in the liver is due to an excess of carbs (glucose) received by the larvae and not an excess of fat
- 12- L141, a reference is needed there
 - 13- Fig. 5b'' and c'': why is there so much GFP background in the yolk sac?
 - 14- L155/156 this again could be unrelated to HFD and just a consequence of more glucose injected by the larvae in the HFD group. Should be performed again with no egg yolk in the food
 - 15- Does GKA exposure affect glut2 in the liver?
 - 16- L170 how does excess fat increase insulin production?
 - 17- Mannoheptulose is a monosaccharide that inhibits all hexokinase (including GCK), so other HK should be evaluated
 - 18- L194 the define Tg(EpRE:RFP) transgenic line (electrophile response element). I believe this line is only used to monitor oxidative stress and not all cellular stress as stated L 196
 - 19- L216 not cellular stress but oxidative stress
 - 20- The first part of the discussion is just a summary of the results not a discussion
 - 21- L247: D-mannoheptulose is not a competitive inhibitor of GCK but of HK
 - 22- L262: other cellular stress not evaluated solely oxidative stress.
- Minor points:
- 1- The grammar of this manuscript should be proof-read. Some sentences are using the wrong tense: L 26-27; L 210
 - 2- please decide between GCK and Gck
 - 3- L256 replace beta but the corresponding Greek letter

Reviewer #3

(Remarks to the Author)

In this manuscript, the authors have studied zebrafish glucokinase which catalyzes the phosphorylation of glucose to glucose-6-phosphate, and shows expression in the liver and pancreas. The authors have analyzed the expression of the three glucokinase isoforms in the different pancreatic cells and in the liver by qPCR and in situ hybridization, and generated a gck:GFP reporter line that can recapitulate their previous findings. In a diabetes model using the pdx1 mutant, glucokinase expression was reported to be reduced and could not be increased by feeding. Activation of glucokinase by dorzagliatin reduced hyperglycemia in the pdx1 mutants.

The manuscript reports novel findings on glucokinase regulation in zebrafish and its regulation under diabetic conditions. The data seem interesting, solid and are mostly statistically proven. However, I also have a couple of concerns that should be addressed by the authors.

Fig. 1: what is the number of biological replicates shown in the heat map? What is the functional relevance of the different isoforms? This has not been addressed in the manuscript.

Fig. 3: why are not all beta cells positive for glucokinase?

Fig. 4: sometimes the yolk is GFP positive, sometimes not. Any idea why?

Fig. 5: in "d" it is shown that the pdx1 mutant has a reduced GFP intensity, but in "e" the effects are lost. Any idea why? Also, the activity measurements in "g" do not seem to correlate with the expression of glucokinase? What is shown in "h"? I can't find it in the legend.

Fig. 7: The lowered glucose concentrations in the pdx1 mutant after dorzagliatin treatment are impressive, but it remains conceptually open why? Since glucose must enter the cells before it can be metabolized by glucokinase, I suggest that the authors provide some experimental data that can explain their findings (in addition to their glucokinase activity data). In the M&M section: how were the ROS measurements performed?

Version 2:

Reviewer comments:

Reviewer #2

(Remarks to the Author)

The authors have adequately answered my queries/comments. I believe that now this manuscript is suitable for publication in Communications Biology

Reviewer #3

(Remarks to the Author)

The authors have addressed most of my concerns from the original manuscript. I'm still puzzled by the information about the biological replicates used in Figure 1. Three biological replicates for OMIC approaches are too low, and duplicates as

shown for liver do not allow for any conclusions. Therefore, the authors need to include the number of replicates in the figure legends to inform the readers about the weak robustness of the statistical analysis, and secondly, they need to tone down their conclusions such as "exclusively expressed", "no expression", "main", because these statements are not supported by the statistics.

Response to Reviewer 1

We thank the Reviewer for taking the time to assess the manuscript and provide us with relevant and detailed feedback. Herein we have tried to address the concerns that were raised.

1. *Fig 1- It is a little unclear where the data come from for Fig. 1. I am assuming the data presented were from archived data from the paper cited and analyzed by the authors to generate the figure. It would be helpful if this was more clearly stated in the text. Also, differences in zebrafish strains used for these previous studies should be noted.*

We thank the Reviewer for pointing this out and added the data sources in the text in the Methods section (Lines 347-350). Furthermore, we added a more detailed description of the bioinformatic analyses to the Supplementary Material. Additionally, we note that RNA sequencing data comes from the AB fish strain (Line 350), while we use a different genetic background, which we also state now in the Methods section (Lines 327-329).

2. *line 115: "liver anlage"- I am not sure what this refers to. I am not an anatomist, but I could not find any ref to this elsewhere.*

We thank the Reviewer for pointing this out. We changed the term to “developing liver”.

3. *line 121 "We attribute the absence of islet expression to probe penetration problems". Although this is a logical assumption, without any supporting data, you really cannot attribute it to anything. I would change the wording to something like "lack of islet expression may be a result of low probe penetration".*

We agree with the Reviewer and changed the phrasing to: “The lack of islet expression may be a result of low probe penetration into the densely packed endocrine islet.”

4. *"wild-type" does this mean you are using an outbred set of fish? If not you need to say what wild-type line you were using.*

We appreciate the Reviewer for this suggestion and added a clarification in the methods section (Lines 329-330) and refer to *pdx1^{+/+}* controls instead of “wild types” throughout the text and figures to be more accurate.

5. *For each experiment the transgenics and their appropriate controls need to be discussed. For example, in Fig 5/ 6 it was not clear whether the animals were the *pdx*^{-/-}; *gck*:GFP or if they were all *pdx*^{-/-}; *gck*:GFP;*ins*:*dsRED*. For clarity, it would be good to clearly identify the genetic lines for each figure.*

We thank the Reviewer for pointing this out. For clarification we added more labels and enhanced their visibility in Figure 5 and 6.

Response to Reviewer 2

We thank the Reviewer for taking the time to assess the manuscript and provide us with relevant and detailed feedback. Herein we have tried to address the concerns that were raised.

1. Is *gck* expression in the islet always constitutive/? Even in adults

We appreciate the Reviewer's question and measured GFP intensity in the islet area of MIN and HFD fed larvae. We show that GFP expression is constitutive in islets and further confirm this via quantitative RT-qPCR on extracted islets of larvae and adult fish fed either a MIN or a HFD diet. Supplementary Figure 5 including representative images and statistics is added to the Supplementary Material and mentioned in the main text (Line 141 to 145).

Supplemental Figure 5: Constitutive expression of *gck* in the pancreatic islets of zebrafish larvae and adults. *gck*:GFP expression in the endocrine islets (outlined) of larvae fed a MIN diet (a) or HFD (b) from 6 to 10 dpf shown as a maximum intensity projection. c) Quantification of GFP signal in the endocrine islet of MIN and HFD fed larvae (N>9). Box plot with whiskers indicating Min to Max. Statistical significance assessed by t-test. d) *Gck* gene expression levels analyzed via RT-qPCR on isolated islets from larvae fed MIN and HFD from 6 to 10 dpf on larval islets (N=5). Scatter and bar plot shows mean with SD. Statistical significance assessed by t-test. e) *Gck* gene expression levels analyzed via RT-qPCR on dissected islets from fasted and fed adult fish (N>3). Scatter and bar plot shows mean with SD. Statistical significance assessed by t-test.

2. The first sentence in the introduction (L39-40) needs a reference

We thank the Reviewer for pointing this out. A reference is added in the revised version.

3. In the introduction the authors should present the hexokinase family mention that CSK is hexokinase 4 and introduce the differences between HK1,2 and HK4 as they later use Mannoheptulose.

We agree with the Reviewer that this point should be addressed. We added it to the introduction (Lines 44-46)

4. L64 define T2D

We thank the Reviewer for pointing this out and changed t2d to type 2 diabetes.

5. L66 replace beta-cells but beta cell number

We thank the Reviewer for pointing this out and made this change.

6. *L68: gck isoforms has not been well described... but what has been described so far?*

We appreciate the Reviewer's comment and we now include in the Introduction what was previously described about gck in the zebrafish (Lines 75-76)

7. *Fig. 1b and c: how was this figure generated? How were the RNA counted? 1C how did the authors discriminate between the three gck isoforms?*

We thank the Reviewer for these questions. To address them, we have expanded information about the bioinformatic analysis in the Supplementary Material to include a detailed description of the mapping and quantification. Additionally, the tables with the data used in the graphs are added as Supplementary Data.

As we now explain, discrimination between the gck isoforms is based on RSEM quantification. RSEM uses a probabilistic expectation-maximization algorithm to estimate transcript abundances and assign multi-mapping reads to the most probable isoform. A detailed statistical framework and its implementation in RSEM are presented in doi:10.1186/1471-2105-12-323. We also inspected the alignment BAM files to verify a lack of reads mapping to the first exon of gck isoform 3, while reads covering that exon as well as the exon 1beta-2 splice junction were clearly identified in pancreatic endocrine samples.

8. *There is no description of the ScRNAseq experiment used to produce the data presented in Fig. 1b*

We appreciate the Reviewer's comment. Neither Fig 1b nor Fig 1c are based on scRNA-seq data, which was wrongly stated in Line 126. Both figures were created from whole transcriptome (bulk) RNASeq data generated from FACS-sorted cell populations, as described in the original publications (Tarifeño-Saldivia *et al.* 2017, Pozo-Morales *et al.* 2023). We mention the usage of published datasets more explicitly in Line 347-350. While data from public scRNA-Seq datasets corroborate the expression pattern of gck in the pancreas, quantification of isoform abundance in scRNA-Seq data is still challenging, as most methods use oligo-d(T) priming and exhibit a strong 3' bias.

9. *What is the carbohydrate source in the HFD experiment?*

Sparos, the producer of the dry food Zebrafeed, reports that carbohydrate content in Zebrafeed is around 4%. Egg yolk powder similarly contains around 4% carbohydrates. We acknowledge that the amount of carbohydrates is low. This is due to the poor ability of fish to utilize dietary carbohydrates as a major energy yielding substrate (doi:10.1017/S0954422414000018). We have added new text to address the composition of the dry food and egg yolk powder (Lines 372-375). We indicate that this as a limitation of the study in the Discussion (Lines 308-310).

10. *Figure 4 is confusing... Were these larvae fed at all? Did these larvae follow the same feeding protocol (feeding start on day 6) as in figure 3? If so were the 6 dpf larvae fed?*

Larvae imaged at 4, 5 and 6 dpf were unfed as stated in the figure legend and in the main text (Line 149). We added an explanation, that feeding was performed from 6 to 10 dpf to the figure legend (Line 595-596).

11. The main difference in the MIN vs HFD diet is the amount of dry food received by the larvae (twice as much in the HFD protocol). As egg yolk is not rich in carbohydrates the difference observed between Figure d and e can be solely attributed to the amount of carbs fed to the larvae in dry food and be totally independent from an HFD (excess fat) regime. The author should repeat this experiment with excess food, not HFD diet to confirm that the increase of GFP localization in the liver is due to an excess of carbs (glucose) received by the larvae and not an excess of fat.

The Reviewer has brought up an interesting issue and we refer here to point 9, while we also repeated the experiments, as suggested. We fed *pdx1* controls either one time dry food, two times dry food, two times dry food and 30mM glucose (to add an additional carbohydrate source) or HFD (as described) per day and monitored *gck:GFP* expression in the liver and confirmed relative *gck* mRNA expression via RT-qPCR. We observed that feeding of HFD increased *gck:GFP* expression in a more consistent and stable manner as compared to feeding of 2 times MIN or 2 times MIN supplemented with 30mM glucose. We added these results as Supplementary Figure 6 and mentioned in the main text (Lines 154-156).

Supplemental Figure 6: Different feedings regimens affect *gck* expression. a) *Gck:GFP* expression in liver (outlined) of larvae fed from 6 to 10 dpf either one time MIN, HFD, two times MIN supplemented with 30mM Glucose or two times MIN per day. b) Quantification of GFP signal in the liver of 10 dpf larvae fed different feeding regimens (N>12). Box plot with whiskers indicating Min to Max. Statistical significance assessed by One-way ANOVA. P-values of multiple comparisons are indicated. c) *Gck* gene expression levels analyzed via RT-qPCR in 10 dpf larvae fed different feeding regimens (N>2). Scatter and bar plot shows mean with SD. Statistical significance assessed by One-way ANOVA. P-values of multiple comparisons are indicated.

As we explain above (point 9), increased feeding, whether dry food or egg yolk, increases the carbohydrates received by the larvae. We use the term ‘high fat diet’ to refer to egg yolk feeding, as previously used in the literature, which provides a source of increased nutrients, including carbohydrates. We agree that carbohydrates are most likely responsible for *gck* regulation. It is difficult to strictly isolate carbohydrate dependency of *gck* regulation, as feeding variability is an unavoidable confounding factor.

12. L141, a reference is needed there

We thank the Reviewer for pointing this out. A reference is added in the revised version.

13. Fig. 5b'' and c'': why is there so much GFP background in the yolk sac?

In 10 dpf larvae the yolk sac is already depleted. The signal observed is autofluorescence in the gut from ingested food. This has also been previously reported by other groups who performed feeding (e.g. <https://doi.org/10.1089/zeb.2017.1440> Figure 3). We added an explanation to line 152-153 and the figure legend (Line 606).

14. L155/156 this again could be unrelated to HFD and just a consequence of more glucose injected by the larvae in the HFD group. Should be performed again with no egg yolk in the food

We refer the Reviewer to point 11 and the newly added Supplemental Figure 6.

15. Does GKA exposure affect glut2 in the liver?

We thank the Reviewer for this interesting question. Unfortunately, there is a limited selection of zebrafish-specific antibodies on the market. We tested a Glut2 antibody from Osenses, but did not detect a specific signal and we were not able to test other Glut 2 antibodies. Nevertheless, we tried to address this question by assessing glucose uptake in the liver in response to Dorzagliatin treatment using the fluorescent glucose analogue 2-NBDG. Additionally, we measured expression levels of genes associated with glucose metabolism, including the glucose transporter *glut2*, via RT-qPCR. We show increased glucose uptake, an upregulation of *glut2* and a trend towards upregulation of genes associated with glycolysis. These data are added as Supplementary Figure 7, reported in Lines 210 to 217 and discussed in Lines 295 to 298.

Supplemental Figure 7: Dorzagliatin affects glucose uptake and glucose metabolism in diabetic *pdx1* mutants. Uptake of the fluorescent glucose analogue 2-NBDG in the liver (outlined) of 10 dpf *pdx1* mutants fed HFD and treated with DMSO (a) or 2 μ M Dorzagliatin (b). c) Quantification of fluorescent signal from 2-NBDG in the liver of DMSO and Dorzagliatin treated 10 dpf *pdx1* mutants (N>5). Box plot with whiskers indicating Min to Max. Statistical significance assessed by t-test. d) Relative expression levels of *gck*, *hk1*, *pklr*, *glut2* and *ins* assessed via RT-qPCR in 10 dpf *pdx1* mutant larvae, HFD fed, and either treated with DMSO or 2 μ M Dorzagliatin (N>6). Scatter and bar plot shows mean with SD. Statistical significance assessed by t-test.

16. L170 *how does excess fat increase insulin production?*

We recognize that this is an important point. We showed that HFD increased *insulin* mRNA. As mentioned before (point 9), HFD, consisting of dry food and egg yolk, is providing proteins and carbohydrates in addition to fat. We have added new text to address the composition of the dry food and egg yolk powder (Lines 372-375), and discuss the distinction between fish and mammals in the importance of different macronutrients in the diet in the discussion (Lines 308-310).

17. *Mannoheptulose is a monosaccharide that inhibits all hexokinase (including GCK), so other HK should be evaluated*

We thank the Reviewer for this interesting point. However, this work is solely focusing on *gck* and we think that assessing other hexokinases is beyond the scope of this paper. We are not aware of a more specific inhibitor that is readily available. We included this issue in the Discussion (Line 285-287).

18. L194 *the define Tg(EpRE:RFP) transgenic line (electrophile response element). I believe this line is only used to monitor oxidative stress and not all cellular stress as stated L 196*

We thank the Reviewer for pointing this out and corrected this throughout the manuscript.

19. L216 *not cellular stress but oxidative stress*

We made this correction.

20. *The first part of the discussion is just a summary of the results not a discussion*

We provide a summary of the results for orientation before placing our results in a larger context. We are following the recommendations of the International Committee of Medical Journal Editors. (Uniform Requirements for Manuscripts Submitted to Biomedical Journals: Writing and Editing for Biomedical Publication. Updated April 2010.)

21. L247: *D-Mannoheptulose is not a competitive inhibitor of GCK but of HK*

We thank the Reviewer for pointing this out and corrected it.

22. L262: *other cellular stress not evaluated solely oxidative stress.*

See point 18.

23. *The grammar of this manuscript should be proof-read. Some sentences are using the wrong tense: L 26-27; L 210*

We thank the Reviewer for pointing this out. We corrected the abstract and reformulated it (Line 27-35). We also corrected inconsistencies in the discussion (Lines 236-238, 242-243, 255-258, 299, 304).

24. *please decide between GCK and Gck*

We thank the Reviewer for pointing this out. We use the accepted formatting conventions for gene and protein symbols and applied organism-specific formatting guidelines as described here: <https://www.biosciencewriters.com/Guidelines-for-Formatting-Gene-and-Protein-Names.aspx>. Using these, we differentiate between human gene and protein, and zebrafish gene and protein, which

leads to 4 different formats. Nevertheless, we found some inconsistencies in the formatting and we corrected them.

25. L256 replace beta but the corresponding Greek letter

We thank the Reviewer for pointing this out and corrected it.

Response to Reviewer #3

We thank the Reviewer for taking the time to assess the manuscript and provide us with relevant and detailed feedback. Herein we have tried to address the concerns that were raised.

1. *Fig. 1: what is the number of biological replicates shown in the heat map? What is the functional relevance of the different isoforms? This has not been addressed in the manuscript.*

We thank the Reviewer for these comments. We added a more detailed description of the bioinformatic analysis and the excel files with the data represented in the graphs to the Supplementary Material. In the original studies, three biological replicates were sequenced for each endocrine cell type and two for each time points in the liver dataset.

Regarding the functional relevance of the different isoforms, we added to the Introduction Lines 46-49:

“In mammals, two organ specific promoters control the expression *GCK*⁷. Differential splicing results in hepatic and endocrine isoforms that are functionally indistinguishable, although there are differences in amino acid sequence at the N-terminal ends of the *GCK* isoforms⁸.”

2. *Fig. 3: why are not all beta cells positive for glucokinase?*

We appreciate this interesting question. This is not zebrafish specific but also observed in the single cell sequencing datasets from human pancreas (<https://cells.ucsc.edu/?ds=human-pancreas>). We can only speculate that this might be because of the reported heterogeneity of the beta cell population. We added this notion to the discussion part (Lines 259-263).

3. *Fig. 4: sometimes the yolk is GFP positive, sometimes not. Any idea why?*

We thank the Reviewer for bringing up this question. In 10dpf larvae the yolk sac is already depleted. The signal observed is autofluorescence in the gut from ingested food. This has also been previously reported by other groups who performed feeding (e.g. <https://doi.org/10.1089/zeb.2017.1440> Figure 3). The degree of autofluorescent signal depends on the respective feeding regime. MIN fed fish show less autofluorescence than HFD fed fish. We added an explanation to line 152-153 and the figure legend (Line 606).

4. *Fig. 5: in “d” it is shown that the *pdx1* mutant has a reduced GFP intensity, but in “e” the effects are lost. Any idea why? Also, the activity measurements in “g” do not seem to correlate with the expression of glucokinase? What is shown in “h”? I can’t find it in the legend.*

We thank the Reviewer for pointing out this mistake. Indeed, a part was missing in the figure legend and labelling got mixed up. Glucokinase expression is shown in g) and activity measurement in h). Expression and activity are both elevated in *pdx1*^{+/+} larvae fed an HFD diet while there is no such response in *pdx1* mutants. We corrected the mistake in the labelling.

With regards to GFP intensity: in MIN fed animals *gck* expression is generally low in *pdx1*^{+/+} controls and in *pdx1* mutants (Figure 5e) because of the limited nutrient supply. Upon HFD feeding, there is increased *gck* expression in controls (compared to MIN fed controls) but very variable expression in *pdx1* mutants. This inconsistency in response could be in part due to the variable severity of the *pdx1* phenotype, as samples show differences in beta cell number and *insulin* expression. Another reason

could be inconsistent uptake of nutrients between larvae, for which we did not control. We hypothesize that these factors contribute to the lack of a significant reduction in GFP intensity in HFD fed mutants compared to controls 10 dpf.

5. Fig. 7: The lowered glucose concentrations in the pdx1 mutant after dorzagliatin treatment are impressive, but it remains conceptually open why? Since glucose must enter the cells before it can be metabolized by glucokinase, I suggest that the authors provide some experimental data that can explain their findings (in addition to their glucokinase activity data).

We are grateful for the Reviewer's interesting comment.

To provide supporting evidence for enhanced glucose uptake and metabolism, we performed glucose uptake experiments using the fluorescent glucose analogue 2-NDBG and performed RT-qPCR to assess expression levels of genes associated with glucose metabolism in larvae treated with the GKA Dorzagliatin. We are able to show increased glucose uptake, increased *glut2* expression and a trend towards increased expression of several genes related to glycolysis. These data are added to the Supplementary Material, Supplementary Figure 7 (also shown in response to Reviewer 2 point 15) and reported in the results (Lines 210-217).

6. In the M&M section: how were the ROS measurements performed?

This is described in the M&M section in the part on *gck:GFP* and *EpRE:RFP* quantification in the liver (Lines 384-387).

Response to Reviewer 3

We thank the Reviewer for taking the time to re-assess the manuscript.

1. I'm still puzzled by the information about the biological replicates used in Figure 1. Three biological replicates for OMIC approaches are too low, and duplicates as shown for liver do not allow for any conclusions. Therefore, the authors need to include the number of replicates in the figure legends to inform the readers about the weak robustness of the statistical analysis, and secondly, they need to tone down their conclusions such as "exclusively expressed", "no expression", "main", because these statements are not supported by the statistics.

We thank the Reviewer for pointing this out. We included the number of biological replicates in the figure legend (Line 563-564). Additionally, we toned down our conclusions and removed or substituted the phrases of overstatement. (Line 98 and 111 – 114).